# Risk Assessment for Heroin Use and Craving Score Using Polygenic Risk Score

**DOI:** 10.3390/jpm11040259

**Published:** 2021-04-01

**Authors:** Chieh-Liang Huang, Ping-Ho Chen, Hsien-Yuan Lane, Ing-Kang Ho, Chia-Min Chung

**Affiliations:** 1Tsaotun Psychiatric Center, Ministry of Health and Welfare, Nan-Tou County 54249, Taiwan; psyche.hcl@gmail.com; 2Ph.D. Program for Aging, College of Medicine, China Medical University, Taichung 40402, Taiwan; ingkangho@gmail.com; 3School of Dentistry, College of Dental Medicine, Kaohsiung Medical University, Kaohsiung 80708, Taiwan; phchen@kmu.edu.tw; 4Institute of Biomedical Sciences, National Sun Yat-sen University, Kaohsiung 80424, Taiwan; 5Department of Psychiatry & Brain Disease Research Center, China Medical University Hospital, Taichung 40402, Taiwan; hylane@gmail.com; 6Department of Psychology, College of Medical and Health Sciences, Asia University, Taichung 41354, Taiwan; 7Graduate Institute of Biomedical Sciences, China Medical University, Taichung 406040, Taiwan; 8Center for Drug Abuse and Addiction, China Medical University Hospital, Taichung 406040, Taiwan

**Keywords:** genetic biomarker, addiction, craving, heroin

## Abstract

Addiction is characterized by drug-craving, compulsive drug-taking, and relapse, and results from the interaction between multiple genetic and environmental factors. Reward pathways play an important role in mediating drug-seeking and drug-taking behaviors, and relapse. The objective of this study was to identify heroin addicts who carry specific genetic variants in their dopaminergic reward systems. A total of 326 heroin-dependent patients undergoing methadone maintenance therapy (MMT) were recruited from the Addiction Center of the China Medical University Hospital. A heroin-use and craving questionnaire was used to evaluate the urge for heroin, the daily or weekly frequency of heroin usage, daily life disturbance, anxiety, and the ability to overcome heroin use. A general linear regression model was used to assess the associations of genetic polymorphisms in one’s dopaminergic reward system with heroin-use and craving scores. Results: The most significant results were obtained for rs2240158 in *GRIN3B* (*p* = 0.021), rs3983721 in *GRIN3A* (*p* = 0.00326), rs2129575 in *TPH2* (*p* = 0.033), rs6583954 in *CYP2C19* (*p* = 0.033), and rs174699 in *COMT* (*p* = 0.036). These were all associated with heroin-using and craving scores with and without adjustments for age, sex, and body mass index. We combined five variants, and the ensuing dose-response effect indicated that heroin-craving scores increased with the numbers of risk alleles (*p* for trend = 0.0008). These findings will likely help us to understand the genetic mechanism of craving, which will help in predicting the risk of relapse in clinical practice and the potential for therapies to target craving in heroin addiction.

## 1. Introduction

Heroin addiction, a chronic relapsing disease characterized by compulsive drug-seeking, drug abuse, tolerance, and physical dependence, is a major public health concern worldwide [1]. It is a complex psychiatric disorder that results from the interaction between biological, environmental, psychological, and drug-related factors. Heroin addiction is strongly influenced by genetic factors, with high heritability estimates of 40% to 60% in different drugs [2,3]. Many genes associated with heroin addiction have been reported using different approaches, suggesting high genetic heterogeneity [2,4]. However, the application of the genetic knowledge to help prevent the occurrence and relapse of heroin addiction and its complications is still lacking.

Previous linkage and association studies of polymorphisms have reported several genetic variants associated with drug abuse, addiction, and related phenotypes [5,6]. These studies include variants in genes encoding the mu and kappa opioid receptors, dopamine receptors D2 and D4, serotonin receptor 1B, GABA receptor subunit gamma 2 [7,8,9], catechol-O-methyltransferase (*COMT*) [10,11], proopiomelanocortin (*POMC*), tryptophan hydroxylase 2 (*TPH2*), and brain-derived neurotrophic factor (*BDNF*) [12]. Physiological and pathological candidate genes may play a role in heroin addiction. The potential mechanism may include drug receptors, neurotransmitters and transporters, drug metabolism enzymes, and the related pathways (e.g., reward modulation, behavioral control, cognitive function, signal transduction, and stress response) [13]. To identify genetic variants involved in heroin addiction, we constructed physiological hypotheses of their function to discover heroin addicts with a high genetic susceptibility. We focused on 13 genes encoding the opioidergic components *OPRK1*, *OPRL1*, *PDYN*, cannabinoid receptor 1 [2], dopaminergic and serotonin components (*COMT)*, *tryptophan hydroxylases 1 (TPH1* and 2 *TPH2*), methadone-metabolizing enzymes (*CYP1A2, CYP2B6,* and *CYP2C19*), and cognitive function-related genes (*GRIN3A*, *GRIN3B*, and *GRM6*). 

Craving is an integral symptom that is considered to be central to the motivational drive in heroin addiction [14]. We explored the genetic basis of craving in order to better understand the addiction process, and to identify potential targets of anti-craving medications and non-pharmacological interventions. We used a craving score as the phenotype to perform a quantitative trait locus (QTL) association study.

## 2. Materials and methods

### 2.1. Patients

The study protocol was reviewed and approved by the Institutional Review Board of China Medical University Hospital (CMUH) (DMR101-IRB1-218) and was in compliance with the Declaration of Helsinki. Informed consent was obtained from all subjects involved in the study. In total, 316 heroin-dependent patients undergoing methadone maintenance therapy (MMT) were recruited from the Addiction Center of the CMUH. As assessed by senior psychiatrists experienced in heroin dependence, all the recruited patients met the criteria from the Diagnostic and Statistical Manual of Mental Disorders, 4th Edition, for heroin abuse and dependence, and had been receiving MMT for at least 6 months at a dosage that had been unchanged for at least 4 weeks at the CMUH. We selected from the extreme margin of the specific phenotype range (e.g., severe heroin addicts receiving MMT) to maximize the power of the study. For each patient, the following clinical information was recorded: sex, weight (in kg), height (in cm), liver function test results, comorbidities, and daily methadone dose.

### 2.2. Heroin-Use and Craving Questionnaire

We designed a 14-item heroin-use and craving (HUC) questionnaire to evaluate the behavior of heroin users undergoing MMT. Each question was scored from 0 to 4. The questionnaire was divided into two parts: Part I (HUC 1–6, possible total scores: 0–24), assessing the urge for heroin and whether the individual can shift their attention away from heroin, and Part II (HUC 7–14, possible total scores: 0–32), investigating the daily or weekly frequency of heroin usage, daily life disturbance, anxiety, and ability to overcome heroin use. In essence, the higher the score, the more severe the heroin craving was. The score of each item was calculated according to the manual.

### 2.3. Candidate Variants Selection and Genotyping

This study focused on the 13 aforementioned genes encoding various metabolic components: *OPRK1*, *OPRL1*, *PDYN*, *CNR1*, *COMT*, *TPH1*, *TPH2*, *CYP1A2, CYP2B6*, *CYP2C19*, *GRIN3A*, *GRIN3B*, and *GRM6*. Single nucleotide polymorphism (SNPs) with minor allele frequency >0.05 were selected on the basis of the previous findings and population data in the SNP databases of the National Center for Biotechnology Information (Han Chinese) in Beijing, China. In total, 44 SNPs were genotyped at the National Center for Genome Medicine, Taiwan, by using Sequenom iPLEX matrix-assisted laser desorption/ionization time-of-flight mass-spectrometry technology. DNA was extracted from 3 to 10 mL of whole blood using a QIAamp DNA Blood Mini Kit (Qiagen, Valencia, CA, USA) according to the manufacturer’s protocol. Duplicate samples were randomly selected for quality control, and the concordance rate was >0.99 for all SNPs assayed.

### 2.4. Statistical Analysis

Each SNP genotype frequency distribution was examined for Hardy-Weinberg equilibrium using the chi-square one degree of freedom goodness-of-fit test. Demographic characteristics and clinical parameters were evaluated with a chi-squared contingency table for categorical variables and a *t* test for continuous variables in both sexes. Single SNP association analyses were conducted with a general linear model, under dominant or recessive model assumptions. We performed QTL mapping for HUC. The general linear model was used to assess associations of genetic polymorphisms in the reward pathway with heroin-use and craving scores, making adjustments for sex, age, and body mass index (BMI). 

## 3. Results

### Patient Characteristics

The demographic details and heroin use history of the subjects are summarized in Table 1. No significant differences were observed between male and female patients with respect to HUC questionnaire score, maximum dose, heroin use duration, education level, and marital status, but significant differences were noted in the mean age, BMI, and heroin onset age (Table 1).

A total of 39 SNPs from 13 genes were not associated with HUC questionnaire scores (Appendix A). Five SNPs (rs2240158 in *GRIN3B*, rs3983721 in *GRIN3A*, rs2129575 in *TPH2*, rs174699 in *COMT* and rs6583954 in CYP2C19) were significantly associated with HUC questionnaire scores. Two SNPs (rs174699 in COMT and rs6583954 in *CYP2C19*) were significantly associated with urge for heroin. Another two SNPs (rs2240158 in *GRIN3B* and rs3983721 in *GRIN3A*) were significantly associated with ability to overcome heroin use (Table 2).

We calculated the genetic risk allele using five significant SNPs from Table 3. The averaged HUC questionnaire score increased with the number of genetic risk alleles in a dose–response manner, as presented in Table 3. Because the HUC questionnaire score increased with the number of risk alleles in an additive or recessive way, we calculated genetic risk allele scores (GRSs) based on their effects on the HUC questionnaire score, assuming the independence of additive risks [15]. The GRSs were generated to quantify the risk alleles for each SNP (example, 2 for homozygous risk alleles, 1 for heterozygous risk alleles, and 0 for the absence of a risk allele). The GRSs were then totaled as defined risk allele scores for each subject. The total scores ranged from 0 to 6 for five SNPs.

In order to evaluate whether there is a combined effect of these genetic variants, three HUC questionnaires were conducted following two models of inheritance (additive or recessive). We found that five SNPs were significantly associated with HUC questionnaire score, as well as the urge for heroin and the ability to overcome heroin use; in addition, *COMT*—rs174699 had borderline significance (*p* = 0.0529). The genetic risk allele scores (combined gene effects) were significantly associated with HUC questionnaire scores, urges for heroin, and ability to overcome heroin use (*p* < 0.001; Table 4).

## 4. Discussion

Heroin addiction, similar to other substance use disorders, is a complex disorder resulting from the interplay between environment and genetic predisposition [16]. Heroin addiction is a highly genetically heterogeneous disorder, and is associated with many genes in different populations. As such, it is crucial to understand the mechanism of heroin addiction by identifying variations in the neurobiology and pathogenesis of genes. Our study identified variations in five candidate genes, including *GRIN3A*, *GRIN3B*, *CYP2C19*, *TPH2*, and *COMT*, contributing to susceptibility to heroin addiction. All the variants identified are from non-coding regions. As expected for a complex genetic disorder, each variant was shown to have a small effect on the risk of heroin addiction. The *R*-squared ranged from 3% to 5% for each genetic variant. This explained a variance in the combined effects of multiple genes on heroin-using and craving scores, and ability to overcome heroin use, of 6.3% (Table 4).

The variants in the *GRIN3A* and *GRIN3B* genes have been shown to be associated with the development of addictive behaviors [17,18,19]. We found that rs2240158 in *GRIN3B* and rs3983721 in *GRIN3A* were associated with urge for heroin and ability to overcome heroin use. *GRIN3A* and *GRIN3B* each encode a subunit of the N-methyl-D-aspartate (NMDA) receptors, which belong to the superfamily of glutamate-regulated ion channels, and are involved in the physiological and pathological processes of the central nervous system. Some genetic variations in the glutamatergic system have been reported to contribute to vulnerability to drug addiction [17,20]. *GRIN3A* and *GRIN3B* may play roles in heroin addiction, acting as dominant-negative modulators of NMDA receptors.

Several variants in *TPH2* have previously been associated with heroin addiction in African Americans and in Hispanics, and are also associated with phenotypes related to smoking status, neuropsychiatric disorders, higher reward dependence, and personality traits [16,21]. TPH is the main 5-HT-synthesizing enzyme in the biosynthesis of the 5-HT pathway in the brain. In the current study, we found that the variant rs2129575 in TPH2 was associated with heroin-using and craving score, which supports the findings of previous studies. 

Our findings indicate that the *COMT* gene is associated with urge for heroin, which plays a critical role in heroin dependence [10,22]. The *COMT* gene SNP rs174699 was associated with urge for heroin. Individuals with the CC genotype would thus be expected to exhibit a stronger urge for heroin use than individuals with the CT or TT genotypes. *COMT* plays an important role in the regulation of synaptic dopamine levels, and in the reinforcement mechanism of drug dependence. The *COMT* SNP rs174699 was selected as a possible marker of genetic predisposition to addiction.

The *CYP2C19* gene encodes a CYP450 enzyme that contributes to methadone metabolism, dependent on treatment dose, plasma concentration, and the side effects of methadone [23]. Methadone maintenance therapy is an established treatment for heroin dependence. The *CYP2C19* gene SNP rs6583954 was found to be associated with heroin-using and craving score, and the ability to overcome heroin use. Individuals with the T allele would thus be expected to exhibit a stronger ability to overcome heroin use than individuals with the C allele. Variations in the *CYP2C19* gene may influence the steady-state plasma concentrations of methadone and the plasma concentrations of methadone [24], while the SNP rs6583954 in the *CYP2C19* gene may potentially serve as an indicator for ability to overcome heroin use.

There are some limitations in this study that need to be considered. The results were compared to those of previous studies. Some variants from previous reports were not genotyped, and the phenotype and population were often different. We cannot determine if the negative findings in this study are true negatives, because the SNP coverage of some of the genes may be insufficient to exclude real associations. Based on known or inferred biologically functional genes, we used a candidate gene approach that allows for scanning a limited number of SNPs and is limited by current knowledge. Our study had only limited phenotypic and genotypic information. It is thus necessary to genotype more novel susceptibility genes identified by genome-wide association studies (GWAS) [20,25,26,27]. Secondly, the sample size was small compared to those in many other studies. We performed a quantitative trait locus association study to identify genetic variants affecting the HUC questionnaire score. Without a normal control in this study, we have suggested that larger-scale studies are warranted in order to further confirm this association, and to use it practically in the population without genetic risk.

## 5. Conclusions

This study suggests an extension to the list of susceptibility genes and variants underlying heroin addiction. Although the variants identified in this study are suggested for assessing the degree of urge for heroin and the ability to overcome heroin use, they may uncover a genetic basis for craving and heroin-addictive behavior. These findings will likely help us understand the genetic mechanism of craving, which will be helpful in predicting the risk of relapse in clinical practice as well as the potential applicability of therapies for targeting craving in heroin addiction.

## Figures and Tables

**Table 1 jpm-11-00259-t001:** Subject characteristics.

	Male (*n* = 259)	Female (*n* = 67)	
Variables	Mean (*SD*)	Mean (*SD*)	*p*-Value
Age	43.2 (7.2)	37.8 (6.3)	<0.0001
BMI	23.0 (2.8)	21.6 (2.8)	0.0005
Heroin-using and craving score	27.2 (9.4)	27.9 (11.8)	0.6247
Urge for heroin	10.6 (4.5)	10.7 (5.4)	0.8429
Ability to overcome heroin use	16.7 (5.8)	17.3 (7.0)	0.4946
MMT max dose	73.9 (29.3)	75.2 (34.7)	0.7479
Heroin onset age (year)	25.6 (7.4)	23.2 (6.1)	0.0138
Heroin use duration (year)	8.6 (5.9)	8.4 (5.8)	0.8751
Education level, *n* (%)			
Elementary school or less	18 (7.0)	5 (7.5)	0.0682
Junior high school	124 (47.8)	21 (31.3)	
Senior high school	117 (45.2)	41 (61.2)	
Marital status, *n* (%)			
Never-married	151 (58.2)	39 (58.2)	0.9447
Married	54 (20.9)	15 (22.4)	
Divorced	54 (20.9)	13 (19.4)	

BMI: body mass index; MMT: methadone maintenance therapy.

**Table 2 jpm-11-00259-t002:** Significant associations from single nucleotide polymorphism (SNP) analysis.

				Heroin-Using and Craving Score		Urge for Heroin		Ability to Overcome Heroin Use	
SNP	Allele	Gene	Gene Pathway	*p*-Value *	*p*-Value ^#^	*p*-Value *	*p*-Value ^#^	*p*-Value *	*p*-Value ^#^
rs2240158	T > C	GRIN3B	Cognitive function	**0.02318**	**0.0353**	0.08232	0.1102	**0.02103**	**0.0288**
rs3983721	C > T	GRIN3A	Cognitive function	**0.00945**	**0.0109**	0.074	0.105	**0.00326**	**0.0026**
rs6583954	T > C	CYP2C19	Methadone-metabolizing enzymes	0.05439	**0.033**	0.10597	**0.0387**	0.05517	0.055
rs2129575	G > T	TPH2	Dopamine and serotonin pathway	**0.03343**	**0.0474**	0.05188	0.056	0.06226	0.0948
rs174699	C > T	COMT	Dopamine and serotonin pathway	0.10576	0.069	0.08398	**0.0369**	0.16704	0.1486

* General linear model for association between heroin-using and craving score and genetic SNPs without adjustment for covariates. ^#^ General linear model for association between heroin-using and craving score and genetic SNPs with adjustment for age, body mass index (BMI), and sex. Bold values denote statistical significance at the *p* < 0.05.

**Table 3 jpm-11-00259-t003:** Averaged heroin-using and craving score stratified according to genotypes of variants in reward pathway genes.

Gene	SNP	Risk Score	Genotypes	Heroin-Using and Craving Score	Urge for Heroin	Ability to Overcome Heroin Use
GRIN3B	rs2240158	1	CC	26.97 (9.99)	10.51 (4.64)	16.47 (6.12)
		1	TC	27.84 (9.1)	10.77 (4.31)	17.07 (5.63)
		0	TT	17.14 (10.12)	6.71 (5.15)	10.43 (5.62)
GRIN3A	rs3983721	0	CC	26.46 (9.51)	10.22 (4.69)	16.29 (5.56)
		0	CT	25.37 (9.88)	9.98 (4.51)	15.39 (6.05)
		1	TT	30.64 (9.67)	11.78 (4.47)	18.98 (6.3)
CYP2C19	rs6583954	0	CC	25.35 (10.22)	9.97 (4.75)	15.45 (6.39)
		1	TC	27.63 (9.6)	10.61 (4.59)	17.02 (5.64)
		2	TT	30 (9.72)	12.04 (4.34)	18.04 (6.17)
TPH2	rs2129575	0	GG	26.63 (9.86)	10.46 (4.75)	16.23 (5.56)
		0	GT	25.62 (10.04)	9.89 (4.65)	15.79 (6.32)
		1	TT	29.57 (9.55)	11.62 (4.43)	17.95 (5.9)
COMT	rs174699	1	CC	29.48 (10.25)	11.76 (4.63)	17.83 (6.43)
		0	CT	26.5 (9.18)	10.24 (4.33)	16.31 (5.62)
		0	TT	25.82 (10.57)	10 (4.98)	15.82 (6.35)

**Table 4 jpm-11-00259-t004:** Significant associations—combined effect of multiple genes in the different genetic modes. * Genetic risk alleles were generated from five significant variants.

			Genotypes	Heroin-Using and Craving score		Urge for Heroin		Ability to Overcome Heroin Use	
Gene	Variants	Genetic Mode	(Risk Score)	*R* ^2^	*p*-Value	*R* ^2^	*p*-Value	*R* ^2^	*p*-Value
GRIN3B	rs2240158	Recessive	T(0) vs. C+TC(1)	0.042333	**0.0107**	0.034026	**0.0358**	0.04399	**0.0092**
GRIN3A	rs3983721	Recessive	T(0) vs. C+CT(1)	0.042856	**0.005**	0.026848	**0.04**	0.052769	**0.0017**
CYP2C19	rs6583954	Additive	CC(0)	0.039135	**0.0094**	0.033026	**0.0156**	0.038139	**0.0159**
TPH2	rs2129575	Recessive	T(0) vs. G+GT(1)	0.035141	**0.02**	0.029815	**0.0241**	0.03363	**0.032**
COMT	rs174699	Recessive	C(0) vs. T+CT(1)	0.03043	**0.0308**	0.032585	**0.0152**	0.027691	0.0529
Combined effects	Genetic risk alleles *	Additive	0–6	0.061242	**0.001**	0.046914	**0.0071**	0.063081	**0.0008**

Bold values denote statistical significance at the *p* < 0.05.

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
