# Peer review of "Risk Assessment for Heroin Use and Craving Score Using Polygenic Risk Score"

_jpm, 2021, doi:10.3390/jpm11040259_

Round 1

Reviewer 1 Report

Some grammar and spelling mistakes can be found in the manuscript and the English language should be improved. 

Author Response

1. Some grammar and spelling mistakes can be found in the manuscript and the English language should be improved. 

Authors’ Response:

Thank you for your comment. We revise comments of reviewers. We send the manuscript to English editing to improve grammar and correct spelling mistakes.

Please see  attached file of the certificate of English editing  follows.

Reviewer 2 Report

A general linear regression model is used to asses associations between genetic polymorphisms in reward system and heroin use and craving score.

Line 39-40 rewrite the sentence.

Line 54-57: there are two sentences with two repetitions each of the words "heroin addiction". That makes 4 repetition in 3 lines.

Line 58: have been reported -> have reported

Line 60: These variant in the genes... seems like an unfinished sentence. If the authors would like just to enumerate some of the variants the sentence should read something like: These studies include variants in genes encoding.......

Line 65-66: bad use of parenthesis

Line 66-69: To identify genetic...to heroin adiction 

Line 135: closing parenthesis missing.

Table 2. CYP2C1 should read CYP2C19

Two of the associations (COMT and CYP2C1) are significant only after applying the adjustment for age, BMI and sex. Thus, it should be very clear how these adjustment have been done.

It could be convenient to highlight the significant P-value (using bold, italics or any other way)

Table 3: formatting of the header. Irregular word spacing makes the names of the columns confusing.

Table 3 shows all the SNPs in the 5 genes from table 2. 

The order of genes in table 3 should match those in table 2.

CYP2C1 shor read CYP2C19

141-143: The mean....Table 3. Needs rewording

143: We calculated SNP genotypes score -> SNP genotype scores

144 and following: why these genotypes have a risk score of 1 while the other genotypes have a score of 0 should be explained in a more detailed way instead of just citing the paper by Mealiffe et. al. These results are the key features that support the paper.

160-161: Rewrite the sentence: "The genetic susceptibility of heroin addiction becomes addicted to heroin because of differences in genetic factors" 

Table 4: CYP2C1 should read CYP2C19

The content of the table looks messy mainly because of word spacin in some cells, specially in the TPH2 and last rows.

The paper is limited to a small number of of SNPs and candidate genes and a limited number of samples. More studies are needed to reach more conclusive answers although the identification of this small number of SNPs may deserve publication as they provide evidence supporting their association to different aspecs of heroin addiction. 

Author Response

A general linear regression model is used to asses associations between genetic polymorphisms in reward system and heroin use and craving score.

  1. Line 39-40 rewrite the sentence.

Authors’ Response:

Thank you. We have rewritten the sentence.

  1. Line 54-57: there are two sentences with two repetitions each of the words "heroin addiction". That makes 4 repetition in 3 lines.

Authors’ Response:

Thank you for your comment. We have corrected these errors. Please see Line 119-122, page4

  1. Line 58: have been reported -> have reported

Authors’ Response:

Thank you for your comment. We have corrected it. Please see Line 123, page4.

  1. Line 60: These variant in the genes... seems like an unfinished sentence. If the authors would like just to enumerate some of the variants the sentence should read something like: These studies include variants in genes encoding.......

Authors’ Response:

Thank you for your comment. We have revised as your suggestion. Please see Line125, page4.

  1. Line 65-66: bad use of parenthesis

Authors’ Response:

Thank you. We have corrected it..

  1. Line 66-69: To identify genetic...to heroin addiction

Authors’ Response:

Thank you. We have rewritten the sentence. Please see Line 152, page 5.

  1. Line 135: closing parenthesis missing.

Authors’ Response:

Thank you. We have corrected it.

  1. Table 2. CYP2C1 should read CYP2C19

Authors’ Response:

Thank you. We have corrected it.

  1. Two of the associations (COMT and CYP2C1) are significant only after applying the adjustment for age, BMI and sex. Thus, it should be very clear how these adjustment have been done.

Authors’ Response: Thank you for your comment. Because age, BMI and sex may be a confounder for heroin addiction, we included genes, age, BMI and sex in the same statistical model. We see whether genes were associated with heroin use and craving score independently.

  1. It could be convenient to highlight the significant P-value (using bold, italics or any other way)

Authors’ Response:

Thank you for your comment. We have highlighted the significant P-value

  1. Table 3: formatting of the header. Irregular word spacing makes the names of the columns confusing.

Authors’ Response:

Thank you. We have corrected it.

  1. Table 3 shows all the SNPs in the 5 genes from table 2.The order of genes in table 3 should match those in table 2.

Authors’ Response:

Thank you for your comment. We have changed order of genes in table2. The order of genes is matched in Table2-4.

  1. CYP2C1 shor read CYP2C19

Authors’ Response:

Thank you. We have corrected it.

  1. 141-143: The mean....Table 3. Needs rewording

Authors’ Response:

Thank you. We have corrected it.

  1. 143: We calculated SNP genotypes score -> SNP genotype scores

Authors’ Response:

Thank you. We have corrected it.

  1. 144 and following: why these genotypes have a risk score of 1 while the other genotypes have a score of 0 should be explained in a more detailed way instead of just citing the paper by Mealiffe et. al. These results are the key features that support the paper.

Authors’ Response:

Thank you for your comment. We explained genetic risk score calculation as follows

“Because the HUC questionnaire score increased with the number risk alleles in an additive or recessive way, we calculated genetic risk allele scores (GRSs) based on their effects on the HUC questionnaire score, assuming the independence of additive risks 15. The GRSs were generated to quantify the risk alleles for each SNP (example, 2 for homozygous risk alleles, 1 for heterozygous risk alleles, and 0 for the absence of a risk allele). The GRSs were then totaled as defined risk allele scores for each subject. The total scores ranged from 0 to 6 for five SNPs.”. Please see line 274-281, page9.

  1. 160-161: Rewrite the sentence: "The genetic susceptibility of heroin addiction becomes addicted to heroin because of differences in genetic factors"

Authors’ Response:

Thank you for your comment. We have revised the sentence as follows.” Heroin addiction is a highly genetically heterogeneous disorder, and is associated with many genes in different populations.” Please see line 333, page10.

  1. Table 4: CYP2C1 should read CYP2C19

Authors’ Response:

Thank you. We have corrected it.

  1. The content of the table looks messy mainly because of word spacin in some cells, specially in the TPH2 and last rows.

Authors’ Response:

Thank you for your comment. We have revised table4 and made it clear. Please see table 4.

The paper is limited to a small number of of SNPs and candidate genes and a limited number of samples. More studies are needed to reach more conclusive answers although the identification of this small number of SNPs may deserve publication as they provide evidence supporting their association to different aspecs of heroin addiction. 

This manuscript is a resubmission of an earlier submission. The following is a list of the peer review reports and author responses from that submission.